# Evidence of p75 Neurotrophin Receptor Involvement in the Central Nervous System Pathogenesis of Classical Scrapie in Sheep and a Transgenic Mouse Model

**DOI:** 10.3390/ijms22052714

**Published:** 2021-03-08

**Authors:** Tomás Barrio, Enric Vidal, Marina Betancor, Alicia Otero, Inmaculada Martín-Burriel, Marta Monzón, Eva Monleón, Martí Pumarola, Juan José Badiola, Rosa Bolea

**Affiliations:** 1Centro de Encefalopatías y Enfermedades Transmisibles Emergentes, Facultad de Veterinaria, Instituto Agroalimentario de Aragón - IA2 (Universidad de Zaragoza - CITA), 50013 Zaragoza, Spain; tbarrio@unizar.es (T.B.); mbetancorcaro@gmail.com (M.B.); aliciaogar@unizar.es (A.O.); mmonzon@unizar.es (M.M.); emonleon@unizar.es (E.M.); badiola@unizar.es (J.J.B.); 2Centre de Recerca en Sanitat Animal, Universitat Autònoma de Barcelona (UAB)-Institut de Recerca i Tecnologia Agroalimentàries, Bellaterra, 08193 Barcelona, Spain; enric.vidal@irta.cat; 3LAGENBIO (Laboratorio de Genética Bioquímica), Facultad de Veterinaria, Instituto Agroalimentario de Aragón - IA2 (Universidad de Zaragoza - CITA), 50013 Zaragoza, Spain; minma@unizar.es; 4Departamento de Medicina y Cirugía Animal, Facultad de Veterinaria, Universitat Autònoma de Barcelona (UAB), Bellaterra, 08193 Barcelona, Spain; marti.pumarola@uab.cat

**Keywords:** prion disease, scrapie, neurotrophin, p75^NTR^, astrocyte, transgenic mice

## Abstract

Neurotrophins constitute a group of growth factor that exerts important functions in the nervous system of vertebrates. They act through two classes of transmembrane receptors: tyrosine-kinase receptors and the p75 neurotrophin receptor (p75^NTR^). The activation of p75^NTR^ can favor cell survival or apoptosis depending on diverse factors. Several studies evidenced a link between p75^NTR^ and the pathogenesis of prion diseases. In this study, we investigated the distribution of several neurotrophins and their receptors, including p75^NTR^, in the brain of naturally scrapie-affected sheep and experimentally infected ovinized transgenic mice and its correlation with other markers of prion disease. No evident changes in infected mice or sheep were observed regarding neurotrophins and their receptors except for the immunohistochemistry against p75^NTR^. Infected mice showed higher abundance of p75^NTR^ immunostained cells than their non-infected counterparts. The astrocytic labeling correlated with other neuropathological alterations of prion disease. Confocal microscopy demonstrated the co-localization of p75^NTR^ and the astrocytic marker GFAP, suggesting an involvement of astrocytes in p75^NTR^-mediated neurodegeneration. In contrast, p75^NTR^ staining in sheep lacked astrocytic labeling. However, digital image analyses revealed increased labeling intensities in preclinical sheep compared with non-infected and terminal sheep in several brain nuclei. This suggests that this receptor is overexpressed in early stages of prion-related neurodegeneration in sheep. Our results confirm a role of p75^NTR^ in the pathogenesis of classical ovine scrapie in both the natural host and in an experimental transgenic mouse model.

## 1. Introduction

Neurotrophins are a group of growth factors that exert important functions in the nervous system of vertebrates. They are synthetized and secreted by neurons and other cell types and regulate critical processes, from neuron maturation and synaptic plasticity to maintenance of the nervous tissue during adulthood [1,2]. Their activity has been linked to a number of neuropathological conditions, and changes in their expression have been observed as a response to cell and tissue damage [3,4].

Neurotrophins constitute a family of structurally and functionally related peptides that include the nerve growth factor (NGF), the brain-derived neurotrophic factor (BDNF), and the neurotrophin 3 (NT-3), among others [2]. All of them are synthesized in the form of pro-neurotrophins, which are later processed intra or extra-cellularly and converted into their mature forms [5,6].

The action of neurotrophins depends on two types of transmembrane receptors: the tyrosine-kinase receptors TrkA, TrkB, and TrkC, and the p75 neurotrophin receptor (p75^NTR^), that belongs to the tumor necrosis factor receptor (TNFR) superfamily [7]. While Trk receptors A, B, and C join to mature forms of NGF, BDNF, and NT-3, respectively, and trigger pro-survival signals, p75^NTR^ respond to both pre-processed and mature forms, although it has higher affinity for pro-neurotrophins [2,6]. In both cases, this receptor activates alternative signaling pathways leading to apoptosis and cell death [1]. Interestingly, p75^NTR^ is also able to interact with other membrane elements, including Trk receptors, in which case it increases their affinity for mature neurotrophins and contributes to pro-survival signaling [8,9], and with sortilin, which participates in the apoptosis outcome through not-yet clarified mechanisms [2,10]. For all these reasons, p75^NTR^ is positioned at the crossroad between cell survival and death and thus has attracted interest due to its potential use as a biomarker for neurodegenerative disorders and, more importantly, as a therapeutic agent.

Prion diseases represent a new paradigm of neurodegenerative conditions. They are caused by unconventional transmissible agents called prions, consisting of a misfolded protein (pathogenic prion protein, or PrP^Sc^) with the ability to transfer its aberrant conformation to the otherwise healthy, physiological cellular prion protein, or PrP^C^ [11,12]. The accumulation of the pathogenic isoform in the brain triggers currently unclarified mechanisms that lead to a neurodegenerative disorder characterized by spongiosis, gliosis, and sometimes presence of amyloid deposits [13,14,15,16,17].

Prions are present in multiple species, including humans and domestic animals. Bovine spongiform encephalopathy (BSE) affects cattle and can be transmitted to humans through consumption of contaminated meat, causing the variant form of Creutzfeldt-Jakob disease (vCJD) [18]. Sheep suffer a form of prion disease called scrapie, which transmits horizontally between individuals. Prion diseases can be experimentally induced in transgenic mice expressing the cellular prion protein (PrP^C^) of other species, by intracerebral inoculation of brain material from infected individuals [19]. These murine models recapitulate most of the features of the natural disorder, and have been routinely used for the study of all aspects of prionopathies [20,21,22].

Studies addressing the relationship between prion pathogenesis and neurotrophins, although scarce, go in line with several pieces of evidence that link these growth factors and neurodegeneration. One example of such a study proved that PrP106-126, a synthetic peptide homologous to the fragment between amino acids 106 and 126 of human PrP, was able to induce apoptosis in an in vitro model (N2a cells) through the activation of p75^NTR^ and the nuclear factor κB (NF-κB); this suggests a direct interaction between these molecules [23,24]. Another study, this time using an in vivo approach, found a positive correlation between the neuropathological hallmarks of prion disease (spongiosis, PrP^Sc^ accumulation and gliosis) and overexpression of p75^NTR^ in a bovinized murine model inoculated with BSE [25]. A third work evidenced a decrease in the levels of p75^NTR^ and other related factors in the brain of scrapie-infected hamsters, which was progressive throughout the course of the disease and correlated with PrP^Sc^ accumulation and neuronal death [26]. These works have prompted the interest on neurotrophins as molecules whose study may shed light on the pathogenesis of prion diseases and other neurodegenerative conditions.

In this study, we analyzed the distribution of three neurotrophins (NGF, BDNF, and NT-3) and four neurotrophin receptors (TrkA, TrkB, TrkC, and p75^NTR^) in brain samples sourced from two different models of scrapie: sheep as the natural host, and Tg338 transgenic mice, which express the ovine PrP^C^, as an experimental model. After a pilot study that confirmed previous observations in other experimental scenarios [25], we focused on the p75 neurotrophin receptor (p75^NTR^) and decided to investigate further its distribution, expression, and correlation with other markers of prion disease.

## 2. Results

### 2.1. Mapping of Neurotrophins and Neurotrophin Receptors in the Brain Reveals Different Staining Patterns 

Two different scrapie models were used: sheep, as the natural host, and Tg338 mice, a transgenic line that expresses ovine PrP^C^ and, therefore, can be experimentally infected with scrapie prions. In the pilot study, two groups of animals of each model were included: the sheep were either healthy or naturally infected with scrapie, and the mice were either inoculated with a scrapie isolate or with physiological saline. 

Three neurotrophins (NGF, BDNF, and NT-3) and four neurotrophin receptors (TrkA, TrkB, TrkC, and p75^NTR^) were mapped in the brain of sheep and Tg338 mice. Each combination of marker and model (ovine or murine) gave a distinctive distribution pattern. However, no clear association with the disease was observed in any case, except for p75^NTR^ in mice.

Both mouse and sheep brains showed neuronal labeling together with neuropil staining of variable intensity for NGF and BDNF (Appendix A). In neuronal bodies, immunostaining was located within the cytoplasm. Staining of other cell types or structures in the brain was not observed, suggesting that the distribution of these markers is restricted to neurons at both the pericarion and their projections through the neuropil. This labeling pattern was ubiquitous in medulla oblongata, midbrain, cortices, and hippocampus of both infected and control mice and in medulla oblongata and hippocampus of sheep. 

The distributions of NGF and BDNF were similar, suggesting similar or parallel roles in the nervous tissue. No evident differences in the immunostaining of any of these markers were detected between infected and control individuals on either sheep or mice.

Similarly, NT-3 presented a low-intensity intracytoplasmic neuronal pattern in mouse brains (Appendix A) together with a diffuse neuropil labeling which was conspicuous in the grey matter of striatum (Appendix A [inset]). The neuronal staining seemed to be higher in infected mice (Appendix A), although differences with control mice were scarce. In sheep, intracytoplasmic neuronal labeling was granular in nature (Appendix A) and more intense than that of mice, while differences between control and infected sheep were subtle or inexistent.

With regard to neurotrophin receptors, TrkA followed a weak and inconstant intraneuronal labeling in both mice (Appendix A) and sheep (Appendix A) and was also detected at the level of neuropil. 

TrkB was located at the level of neuronal membranes (perineuronal labeling) and neuronal prolongations through neuropil in control mice (Appendix A), while in infected mice it was less evident (Appendix A). In contrast, infected and control sheep manifested a mild granular intracytoplasmic staining (Appendix A). Finally, TrkC followed a neuronal intracytoplasmic staining pattern together with a conspicuous labeling of neuronal and glial branching in control mice; these branches were particularly notorious in the molecular layer of cerebellar cortex (Appendix A [inset]). Infected mice showed also strong labeling of Purkinje neurons in the cerebellum (Appendix A [inset]). In sheep, intracytoplasmic labeling was weaker and, in opposition with the situation in mice, Purkinje cells staining was only observed in control, but not in infected sheep (Appendix A [insets]). 

Remarkably, p75 neurotrophin receptor manifested two distinct, clearly differentiable immunostaining patterns. As the rest of analyzed markers, the first pattern was a neuronal intracytoplasmic staining that was observed in all brain areas and in both mice (Figure 1A,B) and sheep (Figure 1E,F). 

The second pattern was only visualized in mice and consisted of star-shaped immunostained cells, different from neurons and likely corresponding to the glia, and whose morphology and distribution resembled those of astrocytes. This labeling was particularly conspicuous in white matter structures, including the corpus callosum (Figure 1C). In addition, these immunostained star-shaped cells were more abundant and fibrous in the brains of infected mice than in control mice (Figure 1D). 

Sheep showed the neuronal intracytoplasmic labeling but not the glial labeling, not even after harsher epitope retrieval protocols were tested. However, in the white matter of sheep, some positive glial cells (probably astrocytes and oligodentrocytes) were detected (Figure 1G). No changes were observed between control and infected sheep.

### 2.2. Glial Immunostaining for p75^NTR^ Is Increased in Terminally Scrapie-Infected Mice

After careful assessment of each of the neurotrophins and receptors, we decided to further analyze the distribution of p75^NTR^. Thus, we performed immunohistochemistry for this receptor on an ampler selection of mouse and sheep brain samples. This included three groups for each model: terminally diseased, preclinical and control, non-infected animals. 

The glial labeling was more intense in terminal mice (Figure 2E,F), in comparison with control mice (Figure 2A,B). This was especially notorious in medulla oblongata, where p75^NTR^-positive star-shaped cells were more abundant and fibrous. This increased glial p75^NTR^ pattern was less evident in the preclinical group (Figure 2C,D). 

The intensity of glial p75^NTR^ labeling was semi-quantitatively evaluated in ten areas of mouse brains. Significant differences were found between control and terminal animals in medulla oblongata (*p* = 0.0187) and mesencephalon (*p* = 0.0067). Significant variability was also found at the striatum (*p* = 0.0020), and although post-hoc pairwise comparison did not disclose significant differences between groups, terminal animals showed clearly increased values (Figure 2G).

### 2.3. Global p75^NTR^ Immunolabeling Is Not Altered by Disease Progression in Mice

In contrast to the semi-quantitative evaluation approach, which quantified exclusively the glial pattern, the global p75^NTR^ immunolabeling was assessed by image analysis. No significant differences were found between terminal, preclinical, and control mice in any area (Appendix A). This suggests that, although higher numbers of p75^NTR^-positive glial cells are present in diseased animals, the overall levels of this receptor seem not to be affected by the disease progression. Rather, this higher abundance of alleged astrocytes is likely to reflect the presence of astrogliosis, a neuropathological feature usually found in prion disease–terminally affected individuals. 

### 2.4. Global p75^NTR^ Immunolabeling Is Increased in Several Brainstem Nuclei of Preclinical Scrapie Infected Sheep

In contrast with mice, p75^NTR^ immunostaining in sheep consisted mainly of an intense intraneuronal, pancytoplasmic labeling, together with a generalized staining of neuropil. However, and in accordance with pilot experiments, sheep brain samples lacked the ramified astrocytic labeling observed in mice.

Careful visual examination identified other staining patterns in sheep brains. For instance, in hippocampus, control sheep showed intraneuronal granular deposits located in one of the poles of the pericarion (Appendix A, panel G) and diffuse, fine punctuate staining in the neuropil, while infected sheep showed coarse granular deposits in neuropil (Appendix A, panels H–I) together with granular accumulations widely spread in the cytoplasm of neurons. However, the observed differences were subtle, and it was not possible to quantify them for statistical purposes.

In addition, no differences in immunostaining were observed between control and infected animals in brainstem or spinal cord, in which an intense intraneuronal pancytoplasmic staining, sometimes granular, was observed (Appendix A, panels A, B, D–F), or in white matter tracts, which showed a strong intracytoplasmic targeting of glial cells probably corresponding to oligodendrocytes (Appendix A, panel C). 

The assessment of global p75^NTR^ staining through image analyses disclosed significant differences in several brain nuclei between infected and control sheep (Figure 3). Most of the differences consisted of higher immunostaining intensity in preclinical animals compared with control, non-infected sheep and with clinical sheep. 

The area where the most significant differences were recorded (*p* = 0.0018) was medulla oblongata. Image analysis of the different nuclei included in this region revealed significant differences in hypoglossal nucleus (between control and preclinical and between preclinical and clinical, *p* = 0.0147) and olivary nuclei (between control and preclinical, *p* = 0.0221).

Significance was also found in comparisons between control and preclinical sheep in pons (*p* = 0.0085) and between preclinical and terminal sheep in mesencephalon (*p* = 0.0176) and diencephalon (*p* = 0.0019). Additionally, significant differences were noted between control and preclinical animals in reticular formation (*p* = 0.0303) and vestibular nucleus (*p* = 0.0103), between control and clinical animals in substantia nigra (*p* = 0.0046), and between preclinical and clinical animals in superior colliculus (*p* = 0.0053), ventral nucleus of the thalamus (*p* = 0.0443), lenticular nucleus (*p* = 0.0120), and caudate nucleus (*p* = 0.0530).

### 2.5. Similarities in the Distribution of p75^NTR^- and GFAP-Immunolabeled Glial Cells Suggest p75^NTR^ Expression in Astrocytes from Mice, but not Sheep

We next wondered whether the glial staining pattern seen in mice corresponded to astrocytes. Presence of reactive astrocytes is a frequent hallmark of prion diseases at advanced stages of neurodegeneration [17].

There was an evident parallelism between the immunostaining for glial fibrillary acidic protein (GFAP), a marker of reactive astrocytes, and for p75^NTR^ in mouse brains. Immunohistochemistry against GFAP (Figure 4C,D) revealed a distribution of astrocytes that closely resembled that of p75^NTR^-positive glial cells (Figure 4A,B). However, p75^NTR^ did not only target astrocytic populations but also neurons, as evidenced by the similarity between the staining patterns of p75^NTR^ and NeuN (neuronal nuclei, a neuron-specific marker) in certain areas such as medulla oblongata (Figure 4E,F).

Importantly, the number of astrocytes targeted for GFAP (Figure 4C) seemed to be higher than that of glial cells targeted for p75^NTR^ (Figure 4A), suggesting that only a subpopulation of astrocytes expresses detectable levels of p75^NTR^ in mouse brain. Similar to p75^NTR^-targeted glial cells, GFAP-targeted astrocytes were more abundant and hypertrophic in infected mice, especially in medulla oblongata of terminal animals.

In sheep, immunohistochemistry against GFAP disclosed GFAP-positive protoplasmic astrocytes in high numbers in the *stratum lacunosum-moleculare* and the polymorphic layer of the dentate gyrus (Figure 5). More fibrous astrocytes were observed in white matter tracts of the *hilus*. Infected sheep had an increased population of GFAP-positive astrocytes in these zones, in comparison with control sheep (Figure 5D–F). 

No correlation was observed between p75^NTR^ (Figure 5A–C) and GFAP immunostainings (Figure 5D–F), in agreement with the lack of the glial pattern in p75^NTR^-immunostained sections from sheep. The reason why ovine astrocytes do not express detectable levels of p75^NTR^ remains unknown.

### 2.6. Glial p75^NTR^ Staining Pattern Correlates Positively with the Severity of Spongiosis, PrP^Sc^ Deposition, and Gliosis in Mice 

The visual approach used to assess the relationship between p75^NTR^ and astrocytes in mice was complemented with a semi-quantitative measurement of the severity of the gliosis in GFAP-immunolabeled sections. In addition, spongiform changes and PrP^Sc^ accumulation in the brain were evaluated on hematoxylin-eosin stained sections and with PET-blot, respectively. 

Significant differences between groups were found in all areas except cerebellar cortex for spongiosis and cerebellar cortex and hippocampus for PrP^Sc^ deposits. Gliosis showed significant differences in medulla oblongata, mesencephalon, hypothalamus, striatum, and septal nuclei (Appendix A), roughly in agreement with what was observed with glial p75^NTR^ labeling. 

Correlation coefficients of these three parameters with glial p75^NTR^ labeling were computed and significant results were found in each case (Figure 6), indicating that the accumulation of the causal agent (PrP^Sc^) and its related neuropathological alterations (spongiosis and gliosis) are linked to the presence of p75^NTR^-positive glial cells. Considering all the aforementioned observations, these glial cells are probably astrocytes. 

### 2.7. Spongiosis and PrP^Sc^ Deposits Do Not Correlate with Global p75^NTR^ Labeling in Mice or Sheep

In contrast with glial p75^NTR^ labeling, no positive correlation was noted between global p75^NTR^ labeling (assessed by image analysis) and spongiosis or PrP^Sc^ deposition in mice. Only a slight correlation was found between total p75^NTR^ and gliosis, which was statistically significant (*p* = 0.0259). However, the low correlation coeficient (r = 0.2133) rendered this observation negligible.

On the other hand, spongiosis and PrP^Sc^ deposits assessed semi-quantitatively in sheep brains showed statistically significant variation between groups (Appendix A), but did not correlate with the global p75^NTR^ immunostaining measured by image analysis (Appendix A). 

### 2.8. Confocal Microscopy Confirms the Relationship between Glial p75^NTR^ Patterns and Astrocytes in Mice, but Not Sheep

Considering that medulla oblongata was the brain area with the most abundant p75^NTR^ labeling and PrP^Sc^ deposits in both mice and sheep, we wondered whether there was a direct, topological relationship between these two parameters. To assess this, we used sheep brain samples and performed confocal microscopy combining the anti-PrP antibody L42 with the anti-p75^NTR^ antibody. 

Immunolabeling for p75^NTR^ disclosed patterns that matched with those observed by immunohistochemistry in sheep, including: perineuronal labeling with neuropil staining (Figure 7A), glial labeling, probably corresponding to oligodendrocytes in the white matter (Figure 7B), and intraneuronal staining with evidence of the neurite branching (Figure 7C). In the samples targeted for p75^NTR^ and PrP, a mild labeling of PrP^Sc^ deposits appeared in some areas, but co-localization with p75^NTR^, although found in some cases, was not the general rule (Figure 7D–F).

Given the apparent labeling of astrocytes by the anti-p75^NTR^ antibody and its positive correlation with the distribution of the astrocyte-specific marker GFAP in mice, we aimed at further confirming the co-localization of these two markers. Therefore, we combined the anti-p75^NTR^ and the anti-GFAP antibodies.

A very clear co-localization was observed in mouse samples (Figure 8A), which confirmed that the p75^NTR^-positive star-shaped cells in mouse brains were indeed astrocytes, as suggested by the observations on immunohistochemistry. In contrast, and also reinforcing previous immunohistochemical findings, no co-localization between GFAP and p75^NTR^ was noted in sheep brain samples (Figure 8B), confirming the absence of p75^NTR^ expression in ovine astrocytes, at least to levels detectable by either immunohistochemistry and confocal microscopy.

## 3. Discussion

Our results indicate that increased numbers of p75^NTR^-positive astrocytes are present in terminal stages of neurodegeneration in a murine model of scrapie. This finding correlates with other neuropathological features of prion disease, including spongiosis, PrP^Sc^ accumulation, and gliosis. In a comparison between control, preclinical, and terminal individuals, all areas with significant differences in glial p75^NTR^ labeling showed also significant differences in these three prion disease biomarkers. These results are in agreement with previous studies that also described glial p75^NTR^ immunostaining in a similar murine model of prion disease, as well as an increase of this labeling in infected individuals that roughly coincides with what we described [25]. These pieces of evidence point to a biologically relevant association between this receptor and prion disease pathogenesis. 

Some studies have proposed an important role for p75^NTR^ in the pathogenesis of prion diseases and other neurodegenerative disorders. Not only is this receptor able to mediate cell death after binding of its natural ligands (pre-processed and mature neurotrophins), as extensively reviewed [1], but also upon in vitro interaction with non-neurotrophic molecules, such as PrP106-126 [23,24] and APP (amyloid protein precursor) [27,28]. Such a direct link between the receptor and peptides involved in the pathogenesis of neurodegeneration prompts the suggestion that a similar mechanism be triggered in vivo by PrP^Sc^ deposits in the brain.

However, few studies address these questions using the mouse as the experimental model. Most investigations on neurotrophin actions are performed in rats [29,30,31], humans [32,33], and non-human primates [34,35]. 

In experimentally scrapie-infected mice, it is well known that prion-related lesions in the brain follow a topographical distribution tightly controlled by a combination of agent strain and factors intrinsic to the host. This has led to the development of strain discrimination methodologies based on the assessment of lesion profiles [36] or PrP^Sc^ accumulation patterns [37,38]. However, the mechanisms controlling or influencing this distribution are unclear. Given the role of p75^NTR^ in triggering cell death, it is possible that differential expression of this receptor among brain areas or cell populations governs at least in part the distribution of prion-related lesions, as already proposed [25].

Our observations confirm that the glial cells targeted for p75^NTR^ that were found increased in diseased mice were astrocytes. As exposed above, a previous work with both wild type and transgenic mice modeling prion disease found increased presence of p75^NTR^-positive astrocytes in terminally diseased animals [25]. Other studies, far from the field of prion diseases, described how astrocytes can both secrete several kinds of neurotrophins [39,40] and act as a target of these molecules through the expression of their receptors, including p75^NTR^ [41,42]. Importantly, according to our results, only a subpopulation of astrocytes seems to express detectable levels of the receptor, as the number of p75^NTR^-positive glial cells was lower than that of GFAP-positive astrocytes.

Some physiological functions of p75^NTR^ in astrocytic populations have been described, for instance, arresting of cell cycle and attenuation of astrocytes proliferation [43,44]. Other authors have demonstrated its involvement in pathological conditions, for example, in p75^NTR^-mediated astrocytosis in astrocyte-induced toxicity to motor neurons in amyotrophic lateral sclerosis (ALS) [45,46,47,48]. In addition, upregulation of the p75^NTR^ receptor has been observed after some types of damage of nervous tissue in vivo [49]. Other glial cell types also upregulate p75^NTR^ expression in several neuropathological conditions, including microglia and oligodendrocytes in multiple sclerosis [50], and Schwann cells and aldynoglia in peripheral nerve injury and during the regeneration process [41,42,51,52]. 

In general, further analyses are needed to understand the involvement of p75^NTR^ in the neuroinflammatory response mediated by reactive astrocytes during prion disease pathogenesis. In any case, our results reinforce the crucial role of astrocytes in the neurodegeneration associated to prion diseases, in agreement with previous results [53].

On the other hand, experiments with the natural model of scrapie are also scarce given the intrinsic drawbacks of working with large animals like sheep. In our study, the natural ovine model of the disease did not show the glial p75^NTR^ immunolabeling found in mice, which suggests the absence of p75^NTR^ in sheep astrocytes. This is not rare, since other studies have failed in finding an association between p75^NTR^ and astrocytes in distinct models of prion disease, such as hamsters [26]. 

Despite the absence of p75^NTR^ astrocytic labeling in sheep, image analysis found an increase in the global p75^NTR^ immunostaining in several brain areas and nuclei of preclinical sheep, in comparison with both control and clinical sheep. This observation suggests that an upregulation of this receptor occurs during initial stages of prion-related neurodegeneration.

Overexpression of p75^NTR^ has also been found in other pathological conditions associated with cell death and neurodegeneration. For example, in Alzheimer’s disease (AD), re-expression of p75^NTR^ was observed in human cortical neurons by immunohistochemistry [54]. The reasons for this increase are unclear. Some authors propose that the re-expression of p75^NTR^ in brains of AD patients is mediated by the upregulation of its ligand, NGF [54], since ligand and receptor are linked by a “feed-forward” relationship [55]. However, in our pilot experiments, the levels of NGF were not noticeably increased in infected individuals. Moreover, levels of BDNF and NT-3, known to bind to p75^NTR^ as well, were also found unaltered. A more in-depth assessment of the expression of these neurotrophic ligands should be done to accept or discard this hypothesis in our particular case. 

In contrast with the preclinical group, animals in the clinical stage showed p75^NTR^ levels similar to those of control animals. A likely explanation for this is the neuronal loss occurring at later stages of prion neurodegeneration. In agreement with this, several authors have described a similar loss of p75^NTR^ neuronal expression in organotypic brain slice cultures from rat [56] and mouse [57] and in cultured canine dorsal root ganglia neurons [58], and attributed this observation to neuronal death. 

Neurotrophins and their receptors, and particularly p75^NTR^, are known to play critical roles in the physiology of the nervous tissue and also to be involved in the pathogenesis of several neurodegenerative disorders, although the underpinnings of this process remain ill-defined. The results presented here aim at shedding light on this issue and provide some new insights into the particular case of prion diseases.

## 4. Materials and Methods 

### 4.1. Sheep

Nine Rasa Aragonesa sheep were included in this study. All of them were homozygous for alanine at codon 136, for arginine at codon 154, and for glutamine at codon 171 (ARQ/ARQ). A single sheep from the control group bore heterozygosity leucine/phenylalanine at codon 141 (141L/F) and another sheep from the terminal group was heterozygous histidine/arginine at position 143 (143H/R). They belonged to three different groups: control, preclinical, and clinical. Clinical animals were diagnosed through assessment of clinical signs and sacrificed in the terminal stage of the disease; diagnosis was later confirmed by detection of PrP^Sc^ in their brains by immunohistochemical procedures. Preclinical animals were identified by immunohistochemical detection of PrP^Sc^ in samples of rectal mucosa-associated lymphoid tissue obtained by rectal biopsy, as previously described [59], and confirmed post-mortem. Control animals were sacrificed without clinical signs at ages similar to those of the other groups and were negative to PrP^Sc^ immunohistochemical detection. All three groups were culled at mean ages of 3 to 4 years old. After sacrifice by intravenous injection of sodium pentobarbital, complete necropsy and sampling was performed. Brains were sectioned and fixed in 10% formalin for at least 48 h before processing.

### 4.2. Mice

The Tg338 line used in this study carries the ovine prion protein gene (*Prnp*) and expresses ovine PrP^C^ (VRQ) under the control of the ovine *Prnp* promoter at levels 8-fold that of sheep brain [60]. A total of 18 animals were included in the study. Six were inoculated with physiological saline (control group) and 12 with a second passage scrapie isolate (i.e., a pool of brain homogenates from Tg338 mice inoculated with a natural case of scrapie). Inoculation was performed by the intracerebral route using a precision syringe and under general anesthesia; after inoculation, they were provided adequate analgesia. Mice were caged together and monitored three times per week. From the scrapie-inoculated mice, 6 (the terminal group) were sacrificed at the end stage of disease with 264 ± 11 dpi (mean ± SEM), and the other 6 (the preclinical group) were culled before the onset of clinical signs at 126 ± 4 dpi. The control group was culled at 420 ± 5 dpi without signs. Sacrifice was done by cervical dislocation under heavy anesthesia. After sacrifice, brains were harvested and fixed in 10% formalin for at least 48 h.

### 4.3. Ethics

All experimental procedures in this study were approved by the Ethics Committee for Animal Testing of the University of Zaragoza (permit numbers PI02/08 and PI19/14, approved on 07/04/2014 and 01/02/2018 respectively) and performed in strict accordance with the recommendations for the care and use of experimental animals and in agreement with national law (RD 1201/05). 

### 4.4. Tissue Processing

Mouse brains were trimmed in four sections for the assessment of prion lesions, as previously described (Fraser and Dickinson, 1968), while sheep samples were processed following an adapted protocol. Tissues were then embedded in paraffin wax and mounted in appropriate cassettes. Four micrometer-thick sections were obtained using a microtome and recovered on slides or nitrocellulose membranes for subsequent analysis.

### 4.5. Immunohistochemistry

Dewaxing and rehydration of the sections was achieved by sequential immersion in graded alcohols. For neurotrophins immunohistochemistry, a heat-induced antigen retrieval step was included by incubating the samples in citrate buffer at 96 ºC for 20 min, while for GFAP, this step was skipped. Endogenous peroxidase activity was blocked using a commercial blocking solution (Dako Agilent, Santa Clara, CA, USA), followed by overnight incubation with primary antibodies raised against NGF (rabbit monoclonal antibody, Sigma-Aldrich, Darmstadt, Germany), BDNF (rabbit monoclonal antibody, Abcam, Cambridge, UK), NT-3 (rabbit polyclonal antibody, Abcam, Cambridge, UK), TrkA (rabbit monoclonal antibody, Abcam, Cambridge, UK), TrkB (rabbit polyclonal antibody, Abcam, Cambridge, UK), TrkC (rabbit polyclonal antibody, Abcam, Cambridge, UK), and p75NTR (rabbit polyclonal antibody, Abcam, Cambridge, UK), or by one-hour incubation with a rabbit polyclonal anti-GFAP antibody (Dako Agilent, Santa Clara, CA, USA). The EnVision+ System (Agilent Dako, Santa Clara, CA, USA) was used as the secondary antibody in all cases, followed by development with the DAB+ System (Agilent Dako, Santa Clara, CA, USA). Counterstaining of the samples was performed with hematoxylin.

### 4.6. Hematoxylin-Eosin (H&E) Staining

Dewaxing and rehydration of preparations by sequential immersion in xylene and graded alcohols was followed by incubation in a hematoxylin solution. After rinsing with tap water, preparations were incubated in acid alcohol (1% acetic acid in a 70% ethanol solution) and then counterstained with an eosin solution. Finally, preparations were dehydrated and mounted prior to visualization under light microscope.

### 4.7. Paraffin-Embedded Tissue Blot

Mouse brain samples were subjected to PET-blot as described elsewhere [61]. Briefly, 4-µm paraffin-embedded brain sections were collected onto a nitrocellulose membrane and dried at 37 °C for 24 h. Membranes were then subjected to dewaxing and rehydration and incubated for 2 h in a solution of proteinase K (250 µg/mL) (Invitrogen, Carlsbad, CA, USA) at 56 °C to completely digest PrP^C^. Denaturation of the remaining PrP^res^ was achieved by incubating the membranes in a solution of guanidine thiocyanate 3M. After blocking the membrane with 0.2% BSA to avoid cross-reactivity, detection was carried out through sequential incubation with the monoclonal anti-PrP antibody Sha31 (1:8000) (SPI-Bio, Sherbrooke, Canada) and a secondary alkaline phosphatase (AP)-conjugated antibody (Agilent Dako, Santa Clara, CA, USA), followed by development with NBT/BCIP (Thermo Scientific, Waltham, MA, USA). Membranes were then washed and dried for 24 h at room temperature.

### 4.8. Semi-Quantitative Assessments

Spongiosis and PrP^Sc^ deposition were evaluated in H&E-stained and PET-blotted or immunostained brain sections from both mice and sheep. Glial p75^NTR^ and GFAP immunostaining were also measured in mouse brains. A semi-quantitative standardized methodology was used by attributing scores from 0 (total absence) to 4 (abundant presence) to each parameter in pre-determined brain areas. The average score for each area was computed as the arithmetic mean of the scores in each experimental group (control, preclinical, and terminal).

### 4.9. Image Analysis

Microphotographs of sheep and mouse brain areas and nuclei were taken using an Axioskop 40 microscope and an Axiocam MRc5 camera (Zeiss, Oberkochen, Germany) and the software AxioVision40 v4.6.3.0 (Zeiss, Oberkochen, Germany). Image analyses were performed as previously described [62]. At least three microphotographs of each area or nuclei were taken. The software ImageJ was used for image analysis. Briefly, microphotographs were subjected to color deconvolution using the algorithm for DAB-stained samples. The mean gray value of the processed images was measured and these values were employed to calculate the optical density (OD) of the pictures applying the formula OD = log (max intensity/mean intensity), where max intensity = 255 for 8-bit images. The average optical density for each area or nucleus was computed as the arithmetic mean of measured optical density values. 

### 4.10. Statistics

Graphical representations and statistical analyses of data were done using the software GraphPad Prism 5. In all cases, Shapiro–Wilk and Kolmogórov–Smirnov tests were applied prior to further analyses, in order to assess the normality of the data, which led to the selection of non-parametric tests. The Kruskal–Wallis test was used to search for differences among control, preclinical, and clinical mice and sheep groups, and was followed by pairwise comparisons using Dunn’s post-hoc test. Finally, correlations between total p75^NTR^ immunolabeling, glial p75^NTR^ immunolabeling, spongiosis, PrP^Sc^ deposition, and astrogliosis were studied using Spearman’s correlation coefficients. 

### 4.11. Confocal Microscopy

Dewaxing, rehydration, and heat-induced antigen retrieval were performed as for immunohistochemistry. To avoid background signal caused by paraffin green autofluorescence, sections were immerged in a solution of Sudan Black B (SBB) for 10 min and protected from light, followed by rinsing with 70% ethanol. Endogenous enzymatic activity was blocked with 10% fetal bovine serum for 60 min, and then, samples were incubated with a mixture of two primary antibodies raised in distinct species (rabbit polyclonal anti-p75^NTR^ antibody (Abcam, Cambridge, UK) and either mouse monoclonal anti-GFAP antibody (Dako Agilent, Santa Clara, CA, USA) or monoclonal mouse anti-PrP antibody L42 (SPI-Bio, Sherbrooke, Canada)), at 4 °C overnight. Next, two fluorophore-conjugated secondary antibodies, emitting at different wavelengths (Alexa Fluor 488 and Alexa Fluor Plus 594, Thermo Scientific, Waltham, MA, USA) were employed to visualize the attachment of each of the primary antibodies. Slides were examined using a LSM 510 equipment (Zeiss, Oberkochen, Germany).

## Figures and Tables

**Figure 1 ijms-22-02714-f001:**
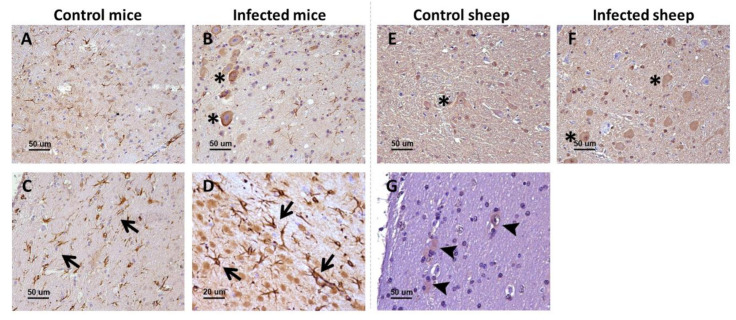
p75^NTR^ receptor in the brain of infected and control mice and sheep. In the upper row (**A**,**B**,**E**,**F**), the general patterns (neuropil staining [not marked] and neuronal intracytoplasmic staining [asterisks]) observed in the two models are displayed, while the lower row (**C**,**D**,**G**) shows specifically the glial staining found in mice (arrows) and sheep (arrowheads). Microphotographs taken from medulla oblongata (**A**,**B**,**D**,**E**,**F**), corpus callosum (**C**), and hippocampus (**G**).

**Figure 2 ijms-22-02714-f002:**
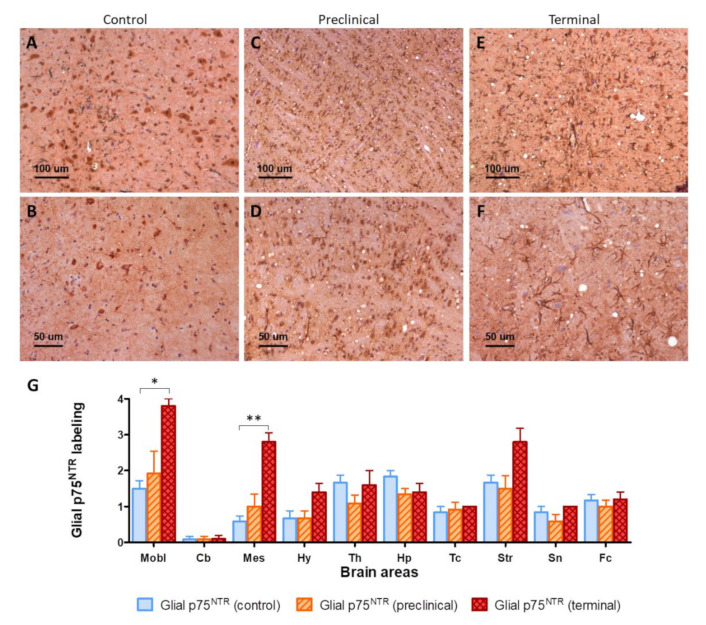
(**A**–**F**) Visual comparison of p75^NTR^ glial immunolabeling between control (**A**,**B**), preclinical (**C**,**D**), and terminal mice (**E**,**F**). Notice the presence of more abundant and fibrous p75^NTR^-positive glial cells (purportedly astrocytes) in infected animals, together with the presence of vacuolization (spongiform lesion) in infected but not in control mice. (**G**) Semi-quantification of glial p75^NTR^ labeling in ten brain areas. Mobl: medulla oblongata, Cb: cerebellar cortex, Mes: mesencephalon, Hy: hypothalamus, Th: thalamus, Hp: hippocampus, Tc: parietal and temporal cortices at the level of thalamus, Str: striatum, Sn: septal nuclei, Fc: frontal cortex. Error bars represent SEM. Kruskal–Wallis test followed by Dunn’s post-hoc pairwise comparison, * *p* ≤ 0.05, ** *p* ≤ 0.01.

**Figure 3 ijms-22-02714-f003:**
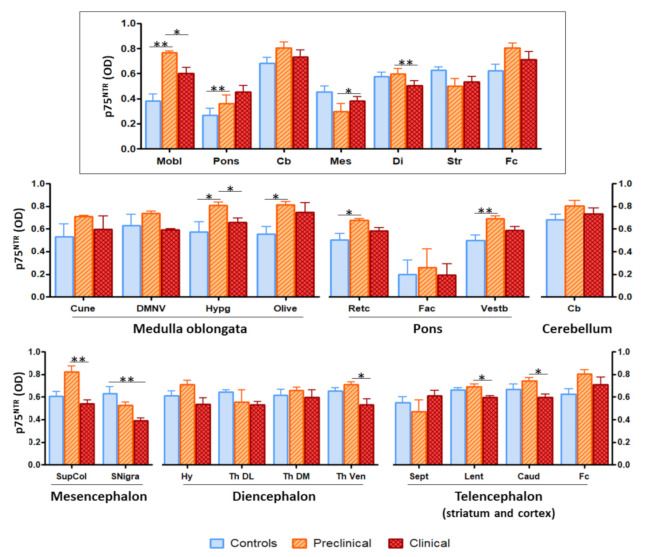
Image analysis of p75^NTR^ immunostaining in sheep brains showed significant differences between groups of animals in several brain areas and nuclei. The upper panel (framed) represents the overall staining in the different brain areas (Mobl: medulla oblongata, Pons: pons, Cb: cerebellum, Mes: mesencephalon, Di: diencephalon, Str: striatum, Fc: frontal cortex), while the lower panels (non-framed), show the staining in specific nuclei within those regions (Cune: cuneate nucleus, DMNV: dorsal motor nucleus of the vagus, Hypg: hypoglossal nucleus, Retc: reticular formation, Fac: facial nucleus, Vestb: vestibular nucleus, SupCol: superior colliculus, SNigra: *substantia nigra*, Hy: hypothalamus, Th DL: dorsolateral nuclei of thalamus, Th DM: dorsomedial nuclei of thalamus, Th Ven: ventral nuclei of thalamus, Sept: septal area, Lent: lenticular nucleus, Caud: caudate nucleus). The height of the bars represents the mean optical density (OD). Error bars represent SEM. Kruskal–Wallis test followed by Dunn’s post-hoc pairwise comparison, * *p* ≤ 0.05, ** *p* ≤ 0.01.

**Figure 4 ijms-22-02714-f004:**
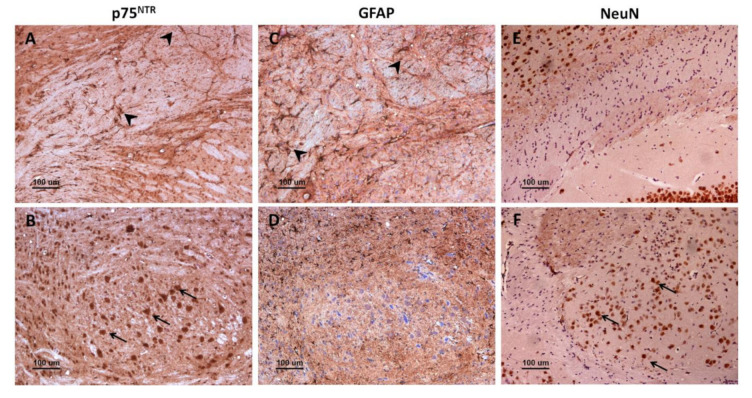
Comparison between p75^NTR^, GFAP, and NeuN immunostaining patterns in mice. Immunohistochemistry against p75^NTR^ (**A**,**B**), GFAP (**C**,**D**), or NeuN (**E**,**F**) in corpus callosum (**A**,**C**,**E**) and in medulla oblongata (**B**,**D**,**F**). Notice that p75^NTR^ labels both glial cells (probably astrocytes) (A, arrowheads) and neurons (**B**, arrows), while GFAP targets astrocytes (**C**, arrowheads) but not other cell types (**D**). Importantly, GFAP-positive astrocytes (**C**, arrowheads) are more abundant than p75^NTR^-positive glial cells (**A**, arrowheads), suggesting that only a subpopulation of astrocytes express p75^NTR^.

**Figure 5 ijms-22-02714-f005:**
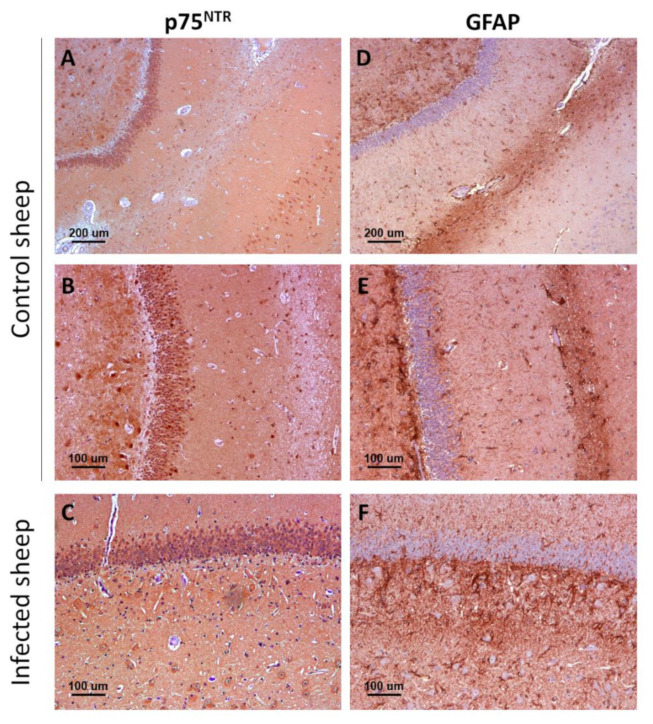
Comparison between p75^NTR^ (**A**–**C**) and GFAP immunostaining (**D**–**F**) patterns in control (**A**,**B**,**D**,**E**) and infected (**C**,**F**) sheep. Notice the higher abundance of GFAP-positive reactive astrocytes (astrogliosis) in infected mice, and the lack of correlation between p75^NTR^ and GFAP distribution patterns, indicative of the fact that ovine astrocytes do not express significant levels of the receptor. Microphotographs taken from hippocampus.

**Figure 6 ijms-22-02714-f006:**
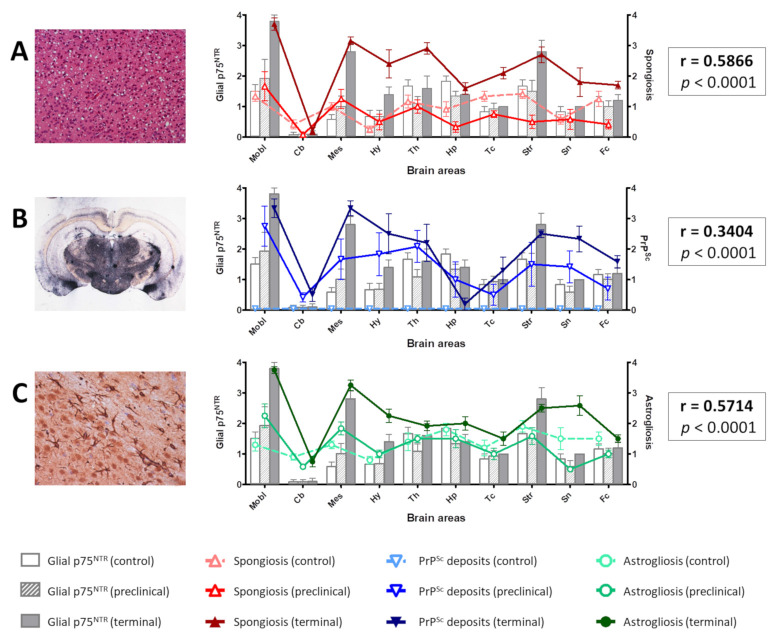
Distribution of spongiform change (**A**), PrP^Sc^ deposits (**B**), and gliosis (**C**) in comparison with the distribution of glial p75^NTR^ labeling in the brain of control, preclinical, and terminal mice. Notice the overlapping of the curves representing the three parameters and the bars representing glial p75^NTR^ staining intensity. Positive Spearman’s correlation coefficients (r) were obtained in each case, which were highly significant (*p* < 0.0001). Mobl: medulla oblongata, Cb: cerebellar cortex, Mes: mesencephalon, Hy: hypothalamus, Th: thalamus, Hp: hippocampus, Tc: parietal and temporal cortices at the level of thalamus, Str: striatum, Sn: septal nuclei, Fc: frontal cortex. Error bars represent SEM.

**Figure 7 ijms-22-02714-f007:**
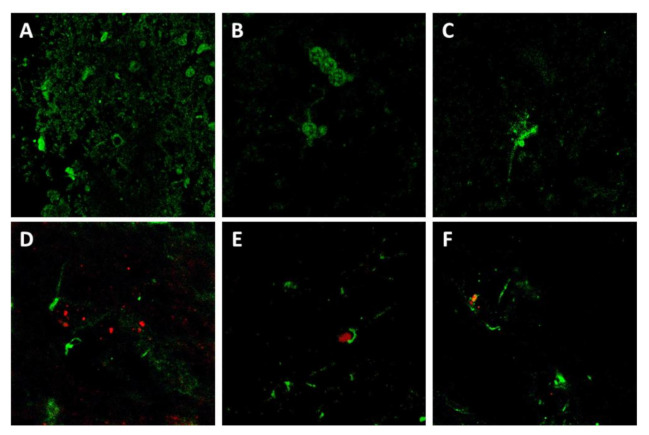
Confocal microscopy for p75NTR (green) and PrP (red) in sheep brain samples. Notice that p75NTR distribution patterns agreed with those observed by IHC, including neuropil and perineuronal staining (**A**), glial staining (probably oligodendrocytes) (**B**), and neuronal intracytoplasmic staining with evidence of neuron processes (**C**). Co-localization of p75NTR with PrP, although observed in several cases (**D**–**F**), was not the general rule.

**Figure 8 ijms-22-02714-f008:**
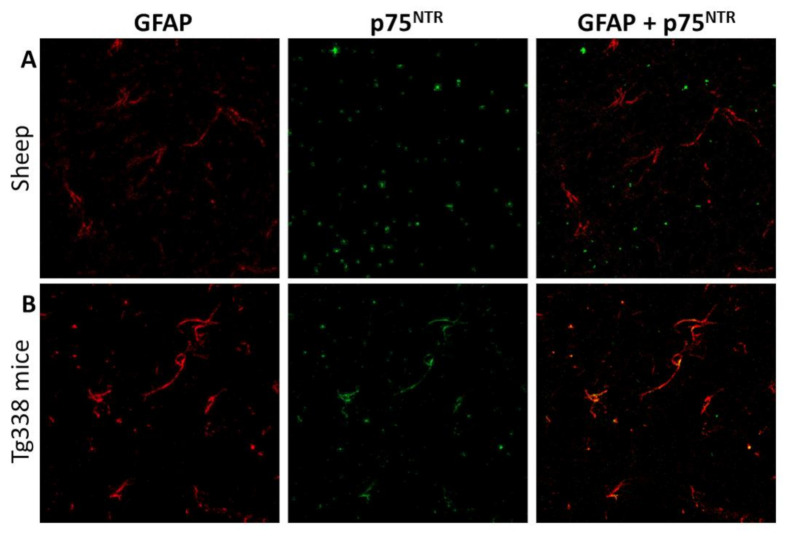
Confocal microscopy for GFAP (red) and p75^NTR^ (green) in sheep (**A**) and mouse (**B**) brain samples. Notice the lack of correlation between both markers in sheep tissues and the clear co-localization that in contrast is observed in mice, suggesting that mouse but not sheep astrocytes express detectable levels of p75^NTR^.

## Data Availability

The data presented in this study are available within the article text, figures and supplementary materials.

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
