# Peer review of "Evidence of p75 Neurotrophin Receptor Involvement in the Central Nervous System Pathogenesis of Classical Scrapie in Sheep and a Transgenic Mouse Model"

_ijms, 2021, doi:10.3390/ijms22052714_

Round 1

Reviewer 1 Report

This study show morphological changes of p75NTR staining pattern of in animal brains during prion disease. In transgenic mouse model, astrocytic p75NTR immunolabelling was increasing but  not in Sheep. However, total p75NTR immunolabeling intensities increased in several brain area of sheep at preclinical stage. These results help for realizing a molecular mechanism of neurodegeneration.

This study is a carefully done. This manuscript is interesting and written well.

However, there is a lack of general information on animal experiment materials.

1. Authors should mention the periods post inoculation for each preclinical and clinical stage.

2. The PrP polymorphism in sheep is important information for experiments of Scrapie prion.

3. Authors should mention information on Primary Antibodies, polyclonal or monoclonal? clone name etc.

4. On Figures 1 and 2, Authors should indicate the degree of PrPSc deposition in the brain of mice and sheep.

5. Figure 4C, D were unclear due to high background staining.

Author Response

Reviewer 1

This study show morphological changes of p75NTR staining pattern of in animal brains during prion disease. In transgenic mouse model, astrocytic p75NTR immunolabelling was increasing but  not in Sheep. However, total p75NTR immunolabeling intensities increased in several brain area of sheep at preclinical stage. These results help for realizing a molecular mechanism of neurodegeneration.

 This study is a carefully done. This manuscript is interesting and written well.

We are thankful for the positive comments about our manuscript.

However, there is a lack of general information on animal experiment materials.

Please find in the following sections our responses to each of your points and suggestions. We highly appreciate the time you dedicated reviewing our manuscript.

  1. Authors should mention the periods post inoculation for each preclinical and clinical stage.

We apologise for this omission. Terminal mice were let alive until the onset of clinical disease with 264±11 dpi (mean ± SEM). Knowing the incubation period, preclinical animals were sacrificed before the onset of clinical signs at 126±4 dpi. The control group was sacrificed at 420±5 dpi without signs. This information has been included in the corresponding section in Material and Methods.

In the case of sheep, this information is not available given that sheep acquired the infection in the field. However all three groups were culled at mean ages of 3 to 4 years old; this has also been stated in the Material and Methods section.

  1. The PrP polymorphism in sheep is important information for experiments of Scrapie prion.

We apologise for this omission. All sheep included in the study were homozygous for alanine at codon 136, for arginine at codon 154 and for glutamine at codon 171 (ARQ/ARQ). A single sheep from the control group bore heterozygosity leucine/phenylalanine at codon 141 (141L/F) and another sheep from the terminal group was heterozygous histidine/arginine at position 143 (143H/R). This information has been included in the appropriate section in Material and Methods.

  1. Authors should mention information on Primary Antibodies, polyclonal or monoclonal? clone name etc.

These data have been included in the corresponding sections in Material and Methods.

  1. On Figures 1 and 2, Authors should indicate the degree of PrPSc deposition in the -brain of mice and sheep.

Animals included in the preliminary study whose brains are presented in Figures 1 and 2 (now moved to Supplementary material) were also included in the final study with p75NTR alone, and thus the information that the reviewer requires can be found in the corresponding sections of the manuscript (2.6 and 2.7) and in Figure 6 (previously Figure 8) and Supplementary Figure S5 (previously Supplementary Figure S3).

  1. Figure 4C, D were unclear due to high background staining.

Background staining in our immunohistochemistry protocol was monitorized by the inclusion of tissue preparations in which the primary antibody incubation step was skipped. We have included a Supplementary Figure (S6) showing this.

Reviewer 2 Report

 The article “Evidence of p75 neurotrophin receptor involvement in the central nervous system pathogenesis of classical scrapie in sheep and a transgenic mouse” by Barrio et al is an interesting subject line. They have tried to dissect the possible role of neurotrophin receptors in the progression of the amyloidal disease using two different model systems. The introduction is well written but still, there is scope to improve. Although authors have mentioned that p75 NTR is associated with several prion conditions but there is no information if there is any relation between TrkA-C with any other known prion condition. Also, the rationale to investigate TrkA-C in prion disease is missing.

Some of the suggestions are mentioned below that need to be addressed to improve the manuscript.

  1. Most of the sections of 2.1 can be shortened by merging. Fig1 and some parts of Fig 2 are negative results and they can be moved to supplements. The result with p75NTR is interesting. It is good to show them with a marked arrow in figure 2M-P. Figure legends are not well written. It is very hard to understand without well-written legends.
  2. Figure 3; control is missing.
  3. Use full name while being used for the first time. The information on how does GFAP is associated with p75NTR or amyloid disease is missing. (see line 246).
  4. Line 271: Fig 7A-C/D-E is not clearly explained. Do these different images represent a different section of the Sheep brain?
  5. Line 353-355 in the discussion section is not clear.
  6. Although there are several lines of evidence that have been presented here regarding the change in abundance of p75NTR in the brain region but a western blot showing the differential expression as confirmatory is missing. It is good to have it.

Author Response

Reviewer 2

The artile “Evidence of p75 neurotrophin receptor involvement in the central nervous system pathogenesis of classical scrapie in sheep and a transgenic mouse” by Barrio et al is an interesting subject line. They have tried to dissect the possible role of neurotrophin receptors in the progression of the amyloidal disease using two different model systems. The introduction is well written but still, there is scope to improve. Although authors have mentioned that p75 NTR is associated with several prion conditions but there is no information if there is any relation between TrkA-C with any other known prion condition. Also, the rationale to investigate TrkA-C in prion disease is missing.

We thank the reviewer for the positive comments. Please find in the following sections our responses to the issues you have raised.

Some of the suggestions are mentioned below that need to be addressed to improve the manuscript.

  1. Most of the sections of 2.1 can be shortened by merging. Fig1 and some parts of Fig 2 are negative results and they can be moved to supplements. The result with p75NTR is interesting. It is good to show them with a marked arrow in figure 2M-P. Figure legends are not well written. It is very hard to understand without well-written legends.

Thanks for your suggestion. We have moved Figure 1 to the supplementary material (Supplementary Figure S1) and have split Figure 2 in two parts: the results for TrkA, TrkB and TrkC have been included in the supplementary material (Supplementary Figure S2), and the results for p75NTR have been moved to a new Figure 1 that also includes the images of the previous Figure 3. All other Figures and Supplementary Figures have been re-numbered, and the references to them have been corrected in the text.

The captions of all Figures and Tables have been modified to be stand-alone for the sake of clarity.

  1. Figure 3; control is missing.

Figure 3 has been eliminated and these images have been included in the new Figure 1.

  1. Use full name while being used for the first time. The information on how does GFAP is associated with p75NTR or amyloid disease is missing. (see line 246).

We have included the meaning for GFAP in the corresponding paragraph (line 273) and in the section abbreviations (line 583) and removed a part of the abstract (line 23) to avoid mentioning several concepts in the abstract before explaining them in the main text.

GFAP is a marker of astrocytes and astrocytosis is a common neuropathological finding in prion diseases as already stated in line 270: “Presence of reactive astrocytes is a frequent hallmark of prion diseases at advanced stages of neurodegeneration [17]”. The association of GFAP with p75NTR is indeed the issue we are investigating in this paper, but there are previous works that described a similar relationship (see Marco-Salazar et al., 2014, ref. [25]).

  1. Line 271: Fig 7A-C/D-E is not clearly explained. Do these different images represent a different section of the Sheep brain?

We apologise for this omission. All these images were taken from the hippocampus of control and infected sheep. This information has been included in the caption of the figure.

  1. Line 353-355 in the discussion section is not clear.

This part of the text refers to other study performed with a similar model (transgenic mice that express bovine PrPC experimentally infected with the agent of bovine spongiform encephalopathy), in which the authors describe similar findings: p75NTR immunostaining is increased in infected animals in comparison with control mice, and a type of glial cells strongly resembling astrocytes were also described that labelled for this protein, suggesting a role of p75 in the astrocyte-mediated pathogenesis in prion diseases.

The paragraph has been modified to insert a brief reference to these results from the Marco-Salazar et al. study: “These results are in agreement with previous studies that also described glial p75NTR immunostaining in a similar murine model of prion disease, as well as an increase of this labelling in infected individuals that roughly coincides with what we described [25]. These pieces of evidence point to a biologically relevant association between this receptor and prion disease pathogenesis” (lines 382-386).

  1. Although there are several lines of evidence that have been presented here regarding the change in abundance of p75NTR in the brain region but a western blot showing the differential expression as confirmatory is missing. It is good to have it.

We agree with the reviewer that a western blot showing the different expression rates of p75NTR in the different groups would be interesting. However, the expression of this protein in the nervous tissue of adult animals is naturally low (see here for mouse: https://www.ncbi.nlm.nih.gov/gene/18053 and here for sheep: https://www.ncbi.nlm.nih.gov/gene/101119064). We performed several attempts to analyse p75NTR expression at the gene (RT-qPCR) and the protein levels (western blot) but all failed due to these low expression rates.  In contrast, the immunohistochemical assessment have the advantage that it allows studying a wide section of the tissue and, thus, its sensitivity is higher than that of other techniques. In addition, our methodology used the EnVision+ system, which enhances the signal produced by the binding of the primary antibody thanks to a large number of secondary antibodies and peroxidase molecules which are directly bound to an activated dextran backbone (up to 100 enzyme molecules and up to 20 antibody molecules per backbone). Moreover, this system is based on HRP-labelled secondary antibodies, and therefore reduces nonspecific staining resulting from endogenous avidin-biotin activity in tissues (see Kämmerer U, Kapp M, Gassel AM, et al. A New Rapid Immunohistochemical Staining Technique Using the EnVision Antibody Complex. Journal of Histochemistry & Cytochemistry. 2001;49(5):623-630. doi: 10.1177/002215540104900509).

Reviewer 3 Report

In this study, the authors attempt to investigate the distribution of NGF, 22 BDNF, NT-3, TrkA, TrkB, TrkC and p75NTR in the brain of scrapie-affected sheep and infected ovinized transgenic mice. They evidenced no changes in neurotrophins except an increase of p75NTR expression just in the mice infected group.  However, the most of marker were “quantified” by observation and this is not a reliable method in research. They should perform the appropriate quantification for each marker

In my opinion, there is an important missing experiment in this manuscript. The authors explained along all the manuscript the importance of glial cells for p75+ cells in the process of this illness, particularly in mice group.

Authors have reported a result of high interest but the results are confusing. They have reported that they are two different pattern of p75 expression in brain mice, corresponding to both neuron and glial cells. Regarding this results there is a relevant finding that maybe the authors did not take into account. They support the theory that both control and infected mice showed an increase of p75 in glial cells by the “observation” of these cells in the tissue. In addition, they have observed this star-shaped cells more abundant in brain mice infect than control.

In my opinion, there are several important point to improve in this manuscript:

  • First, recently have been reported the presence of an unknown glial cells that are positive for p75 neurotrophin receptor (p75NTR): Kim et al, 2019, Ann Clin Transl Neurol, indicating that maybe they have found this kind of cells in both groups (control and infected mice).
  • Authors did not quantify the differences between infected and control mice, just they performed a semi-quantitative analysis based on 0 (absence) or 4 (abundant presence). Taken in mind that these mice were inoculated by using a syringe icv, maybe they are suffering an increase of glial expression induced by a process of tissue invasion. They should have performed a quality analysis for evaluating this situation. Also, the inclusion of a “naïve” group give us more information. In addition, the authors should to include the nº of days passed before they were sacrificed and the nº of days considered for the development of illness.

Other important point is regarding to the 2.5 item of results. 2.5. Similarities in the distribution of p75NTR- and GFAP-immunolabeled glial cells suggest p75NTR expression in astrocytes from mice, but not sheep.

Regarding the results obtained in the figure 6, the conclusions that the authors try to reach are unclear. If the authors would like to compare the distribution of p75NTR in neurons and glial cells, they must be performed two double inmunos: (a) p75NTR + NeuN in order test the expression of   p75NTR in neurons and (b) p75NTR+ GFAP in order to test the expression of p75NTR in glial cells. The authors explained that “In contrast, GFAP did not disclose any neuronal staining (Figure 6D), as did p75NTR (Figure 6B), indicating that GFAP is a specific astrocytic marker and that p75NTR does not only label astrocytes but also other cell populations”. Absolutely GFAP is never going to mark neurons because is a “anti-glial fibrillary acidic protein”

I think, the authors should perform two double immune, in order to clarify this result

OTHER IMPORTANT POINTS:Scale bar are almost invisible and they must be perfectly aligned in the lower left corner

Figure Legend should be more explicated. Please include a reference (*, arrow, etc: as in the figure 2O) indicating that is considered labelling. If they included a magnification, they should indicate the origin of the same (in the original image)

Regarding figure1, the scale bar seems to be different for some images (comparing AD vs 1H OR 1L)

Regarding Figure 2, please include the name of the brain structure in the image

Please include some reference indicating that is considered labelling.

Author Response

Reviewer 3

In this study, the authors attempt to investigate the distribution of NGF, 22 BDNF, NT-3, TrkA, TrkB, TrkC and p75NTR in the brain of scrapie-affected sheep and infected ovinized transgenic mice. They evidenced no changes in neurotrophins except an increase of p75NTR expression just in the mice infected group.  However, the most of marker were “quantified” by observation and this is not a reliable method in research. They should perform the appropriate quantification for each marker.

We understand that immunohistochemistry is a technique that does not allow the quantification of the levels of the studied proteins. However, we did not perform a simple “observation”, as the reviewer states, but a semi-quantification following a previously established scale. Semi-quantitative evaluation of histological preparations has been used in numerous studies on prion diseases as an approach to determine the upregulation of certain proteins. We and others have published papers using this method of evaluation, some of them in this journal. Please find below a list of papers that used this technique in haematoxylin-eosin-stained sections to assess the severity of the spongiform lesion in scrapie-infected mice:

  • Fraser, H.; Dickinson, A.G. The sequential development of the brain lesion of scrapie in three strains of mice. J. Comp. Pathol. 1968, 78, 301-311.

In PrP immunostained sections to analyse the relative abundance of different types of PrPSc accumulation patterns:

  • Gonzalez, L.; Martin, S.; Begara-McGorum, I.; Hunter, N.; Houston, F.; Simmons, M.; Jeffrey, M. Effects of agent strain and host genotype on PrP accumulation in the brain of sheep naturally and experimentally affected with scrapie. J. Comp. Pathol. 2002, 126, 17-29, 10.1053/jcpa.2001.0516 [doi].
  • Gonzalez, L.; Martin, S.; Jeffrey, M. Distinct profiles of PrP(d) immunoreactivity in the brain of scrapie- and BSE-infected sheep: implications for differential cell targeting and PrP processing. J. Gen. Virol. 2003, 84, 1339-1350.
  • Vidal, E.; Acin, C.; Foradada, L.; Monzon, M.; Marquez, M.; Monleon, E.; Pumarola, M.; Badiola, J.J.; Bolea, R. Immunohistochemical characterisation of classical scrapie neuropathology in sheep. J. Comp. Pathol. 2009, 141, 135-146, 10.1016/j.jcpa.2009.04.002 [doi].

And in tissue sections immunostained for different markers:

  • Marco-Salazar, P.; Marquez, M.; Fondevila, D.; Rabanal, R.M.; Torres, J.M.; Pumarola, M.; Vidal, E. Mapping of Neurotrophins and their Receptors in the Adult Mouse Brain and their Role in the Pathogenesis of a Transgenic Murine Model of Bovine Spongiform Encephalopathy. J. Comp. Pathol. 2014, 150, 449-462, 10.1016/j.jcpa.2013.11.209.
  • Otero, A., Bolea, R., Hedman, C. et al. An Amino Acid Substitution Found in Animals with Low Susceptibility to Prion Diseases Confers a Protective Dominant-Negative Effect in Prion-Infected Transgenic Mice. Mol Neurobiol 55, 6182–6192 (2018). https://doi.org/10.1007/s12035-017-0832-8
  • Otero, A., Hedman, C., Fernández-Borges, N. et al. A Single Amino Acid Substitution, Found in Mammals with Low Susceptibility to Prion Diseases, Delays Propagation of Two Prion Strains in Highly Susceptible Transgenic Mouse Models. Mol Neurobiol 56, 6501–6511 (2019). https://doi.org/10.1007/s12035-019-1535-0
  • López-Pérez, Ó., Otero, A., Filali, H. et al. Dysregulation of autophagy in the central nervous system of sheep naturally infected with classical scrapie. Sci Rep 9, 1911 (2019). https://doi.org/10.1038/s41598-019-38500-2
  • Otero, A., Duque Velásquez, C., Johnson, C. et al. Prion protein polymorphisms associated with reduced CWD susceptibility limit peripheral PrPCWD deposition in orally infected white-tailed deer. BMC Vet Res 15, 50 (2019). https://doi.org/10.1186/s12917-019-1794-z
  • López-Pérez, Ó., Toivonen, J.M., Otero, A. et al. Impairment of autophagy in scrapie-infected transgenic mice at the clinical stage. Lab Invest 100, 52–63 (2020). https://doi.org/10.1038/s41374-019-0312-z
  • Guijarro, I.M.; Garcés, M.; Andrés-Benito, P.; Marín, B.; Otero, A.; Barrio, T.; Carmona, M.; Ferrer, I.; Badiola, J.J.; Monzón, M. Assessment of Glial Activation Response in the Progress of Natural Scrapie after Chronic Dexamethasone Treatment. Int. J. Mol. Sci. 2020, 21, 3231. https://doi.org/10.3390/ijms21093231
  • Guijarro, I.M.; Garcés, M.; Marín, B.; Otero, A.; Barrio, T.; Badiola, J.J.; Monzón, M. Neuroimmune Response in Natural Preclinical Scrapie after Dexamethasone Treatment. Int. J. Mol. Sci. 2020, 21, 5779. https://doi.org/10.3390/ijms21165779
  • López-Pérez, Ó.; Bernal-Martín, M.; Hernaiz, A.; Llorens, F.; Betancor, M.; Otero, A.; Toivonen, J.M.; Zaragoza, P.; Zerr, I.; Badiola, J.J.; Bolea, R.; Martín-Burriel, I. BAMBI and CHGA in Prion Diseases: Neuropathological Assessment and Potential Role as Disease Biomarkers. Biomolecules 2020, 10, 706. https://doi.org/10.3390/biom10050706
  • Otero, A.; Betancor, M.; Eraña, H.; Fernández Borges, N.; Lucas, J.J.; Badiola, J.J.; Castilla, J.; Bolea, R. Prion-Associated Neurodegeneration Causes Both Endoplasmic Reticulum Stress and Proteasome Impairment in a Murine Model of Spontaneous Disease. Int. J. Mol. Sci. 2021, 22, 465. https://doi.org/10.3390/ijms22010465
  • Guijarro, I.M.; Garcés, M.; Andrés-Benito, P.; Marín, B.; Otero, A.; Barrio, T.; Carmona, M.; Ferrer, I.; Badiola, J.J.; Monzón, M. Neuroimmune Response Mediated by Cytokines in Natural Scrapie after Chronic Dexamethasone Treatment. Biomolecules 2021, 11, 204. https://doi.org/10.3390/biom11020204

As reflected by this list of our recent publications, our group has extensive experience in using this methodology.

In my opinion, there is an important missing experiment in this manuscript. The authors explained along all the manuscript the importance of glial cells for p75+ cells in the process of this illness, particularly in mice group.

Authors have reported a result of high interest but the results are confusing. They have reported that they are two different pattern of p75 expression in brain mice, corresponding to both neuron and glial cells. Regarding this results there is a relevant finding that maybe the authors did not take into account.

We thank the reviewer for pointing out missing experiments or findings in our manuscript, and we invite him to specify his thoughts about this issue in order to improve our future research involving p75NTR and glial cells.

They support the theory that both control and infected mice showed an increase of p75 in glial cells by the “observation” of these cells in the tissue. In addition, they have observed this star-shaped cells more abundant in brain mice infect than control.

Please refer to previous sections in what concerns the validity of the semi-quantitative evaluation approach.

In my opinion, there are several important point to improve in this manuscript:

  • First, recently have been reported the presence of an unknown glial cells that are positive for p75 neurotrophin receptor (p75NTR): Kim et al, 2019, Ann Clin Transl Neurol, indicating that maybe they have found this kind of cells in both groups (control and infected mice).

We thank to reviewer for the suggestion. The referred paper presents a complex study in which p75 is used as a biomarker to trace certain parts of the pathogenesis of several neurodegenerative diseases, specifically those parts involving Schwann cells. We are aware that p75 is a ubiquitous molecule that is expressed in several types of glial cells, including microglia, oligodendrocytes, Schwann cells and aldynoglia, as stated in lines 419-422. However, the morphology and distribution of p75-positive cells in our study and, more importantly, the results presented in section 2.8 (confocal microscopy for p75 and GFAP) strongly suggest that these cells are astrocytes.

  • Authors did not quantify the differences between infected and control mice, just they performed a semi-quantitative analysis based on 0 (absence) or 4 (abundant presence). Taken in mind that these mice were inoculated by using a syringe icv, maybe they are suffering an increase of glial expression induced by a process of tissue invasion. They should have performed a quality analysis for evaluating this situation. Also, the inclusion of a “naïve” group give us more information. In addition, the authors should to include the nº of days passed before they were sacrificed and the nº of days considered for the development of illness.

See previous comments about the validity of the semi-quantitative approach used in our study.

On the other hand, the possibility that astrogliosis is a consequence of tissue invasion by a concurrent agent is very unlikely because of several reasons:

1) All inoculations were performed in the highest possible aseptic conditions, using disposable needles and needle adapters, and precision syringes that had been thoroughly disinfected with NaOH (for prions) and ethanol 70 % and cleansed with water. All inocula used in intracerebral inoculation were prepared in aseptic conditions and subjected to microbiological analysis prior to inoculation. Immediately after inoculation and during all their lifespan, animals were caged in HEPA-filtered ventilated racks.

2) A cerebral infection resulting from an inappropriate inoculation methodology would have killed the animals in a short period of time. However, animals belonging to the clinical group, which were let alive until the onset of clinical signs, were sacrificed at 264±11 (mean ± SEM). In addition, the preclinical group was sacrificed at 126±4 dpi and the control group at 420±5 with no clinical evidences of intercurrent diseases. We apologise for the omission of this information, which have being included in the appropriate section in Material and Methods.

3) Gliosis is a common and well-described neuropathological hallmark in prion disease, both in natural cases and in experimentally induced disease in rodent models such as ours, as stated in line 67.

Other important point is regarding to the 2.5 item of results. 2.5. Similarities in the distribution of p75NTR- and GFAP-immunolabeled glial cells suggest p75NTR expression in astrocytes from mice, but not sheep.

 Regarding the results obtained in the figure 6, the conclusions that the authors try to reach are unclear. If the authors would like to compare the distribution of p75NTR in neurons and glial cells, they must be performed two double inmunos: (a) p75NTR + NeuN in order test the expression of   p75NTR in neurons and (b) p75NTR+ GFAP in order to test the expression of p75NTR in glial cells. The authors explained that “In contrast, GFAP did not disclose any neuronal staining (Figure 6D), as did p75NTR (Figure 6B), indicating that GFAP is a specific astrocytic marker and that p75NTR does not only label astrocytes but also other cell populations”. Absolutely GFAP is never going to mark neurons because is a “anti-glial fibrillary acidic protein”

 I think, the authors should perform two double immune, in order to clarify this result

We thank the reviewer for the suggestion. We agree that a double immunohistochemistry would be informative to assess the co-localization of p75 with both the astrocytic marker GFAP and the neuronal marker NeuN. However, we have decided to compare the distribution patterns of p75, GFAP and NeuN in single-marker immunostained sections. In order to do so, we performed an immunohistochemistry against NeuN in brain sections from the same animals. As expected, NeuN followed a neuronal pattern that agreed in part with the pattern described for p75, although the latter also marked astrocytes in a fashion comparable to that of the astrocytic marker GFAP. This information has been included in the text (lines 278-280) and illustrated through a revised Figure 4.

We have also removed the sentence stating that GFAP is a specific astrocytic marker given that this information if of general knowledge and not a conclusion obtained from our experiments.

OTHER IMPORTANT POINTS: Scale bar are almost invisible and they must be perfectly aligned in the lower left corner

The scale bars will be sufficiently visible in the full version of the Figures should the article be published.

Figure Legend should be more explicated. Please include a reference (*, arrow, etc: as in the figure 2O) indicating that is considered labelling. If they included a magnification, they should indicate the origin of the same (in the original image)

Figure 2O is now Figure 1E. The glial labelling was already marked with arrows and arrowheads; we have now marked the neuronal staining with asterisks, while the overall neuropil staining is not marked.

There are no magnifications in the Figures, the little boxes are there just to show some particularities of the distribution patterns (e.g. in Supplementary Figure S2, the little boxes in the TrkC row represent the staining pattern of Purkinje cells).

 Regarding figure1, the scale bar seems to be different for some images (comparing AD vs 1H OR 1L)

In Figure 1 (now Supplementary Figure S1), all the images are at the same scale and the scale bars at identical.

In Figure 2 (now Supplementary Figure S2), images A and B were taken with a lower magnification objective and the scale bar represents a different distance (100 um) than in the rest of pictures (50 um). These are the only pictures that we got from the preliminary study (performed in a different laboratory), and in any case was this done to purposefully lead to confusion. In our opinion, however, this does not make the Figure invalid as the intention here is just to show the overall distribution patterns of the different proteins.

Regarding Figure 2, please include the name of the brain structure in the image

This information is already included in the figure caption (now Supplementary Figure S2).

Please include some reference indicating that is considered labelling.

See previous responses to your comment.

Round 2

Reviewer 2 Report

The authors have addressed all concerns raised in my previous report. The manuscript looks much improved and aligned.

Author Response

The authors have addressed all concerns raised in my previous report. The manuscript looks much improved and aligned.

We thank we reviewer for the time she/he spent in reviewing our manuscript and for the positive comments.

Reviewer 3 Report

Authors have responded to some comments and they have tried to improve the MS. However, it is important to explain in the MS, why the control group was sacrificed at 420 dpi versus 264 or 126 dpi of the rest of the groups? Control group should be sacrificed same day post-induction that the terminal group

In the figure 4 there is a mistake: p75NTR is missing

Scale bars shoudl be larger and they must be perfectly aligned in the lower left corner

Author Response

Authors have responded to some comments and they have tried to improve the MS. However, it is important to explain in the MS, why the control group was sacrificed at 420 dpi versus 264 or 126 dpi of the rest of the groups? Control group should be sacrificed same day post-induction that the terminal group

Thanks for your suggestion. Unfortunately we did not have samples from mice sacrificed at the same age at our disposal and we decided to make use of samples obtained from a different experiment, as encouraged by the “Reduction” strategy of animal testing 3 R’s principle. We do not consider that this age difference (156 days, i.e. ~20% of the mouse normal lifespam) is enough to affect our results. In agreement, terminally-diseased mice have even so higher glial p75NTR labelling that their non-infected counterparts, indicating that ageing in not influencing significantly the results obtained for the control group.

In the figure 4 there is a mistake: p75NTR is missing

We apologise for the mistake. It has been corrected.

Scale bars should be larger and they must be perfectly aligned in the lower left corner

This has been corrected in all figures.